# Ecological suitability of Japanese encephalitis virus in Australia: A modelling analysis of vector-host transmission dynamics to potential spillover in humans

Eloise B. Skinner[1‡]*, Benn Sartorius[1‡]*, Luis Furuya-Kanamori[1], Adam T. Craig[1],
Behzad Kiani[1], Brian J. Johnson[2], Kevin T. Moore[3], Roslyn I. Hickson[4,5],
Erin A. Mordecai[6], Gregor Devine[2], Colleen L. Lau[1]

**1** Centre for Clinical Research (UQCCR), Faculty of Health, Medicine and Behavioural Sciences, University of Queensland, Brisbane, Australia, **2** Mosquito Control Laboratory, QIMR-Berghofer Medical Research Institute, Brisbane, Queensland, Australia, **3** School of Environment and Science, Griffith University, Brisbane, QLD, Australia, **4** Commonwealth Scientific and Industrial Research Organisation (CSIRO), Townsville, Australia, **5** Australian Institute of Tropical Health and Medicine, James Cook University, Townsville, Australia, **6** Department of Biology, Stanford University, Stanford, California, United states of America

‡ Joint first authors.
* eloise.skinner@uq.edu.au (EBS); b.sartorius@uq.edu.au (BS)

## Abstract

Japanese encephalitis virus (JEV) is a multi-vector, multi-host pathogen maintained in circulation between *Culex* mosquitoes and waterbirds, with occasional spillover to humans. In Australia, sporadic local transmission of JEV was historically confined to northern Australia until 2021 after which outbreaks occurred for the first time in temperate southern regions in 2022 and 2023, respectively. Following this outbreak, there have been concerns that JEV has potentially become endemic in Australia, posing an ongoing public health risk exacerbated by shifting climatic and environmental factors. We developed and applied a spatially explicit spillover modelling framework which focuses on estimating i) an ecological suitability index for JEV in Australia and, ii) the spillover potential to human populations if endemic transmission is established. To calculate ecological suitability and the potential number of humans that could be exposed, we integrated the ecological and epidemiological conditions that would allow JEV to circulate in Australia and spillover to human populations. An ecological suitability index was calculated by combining the presence of hosts (*Ardeidae* birds, domestic piggeries, feral pigs) and vectors (*Culex annulirostris* and *Culex quinquefasciatus*), host-vector contact rates, and vector infection and transmission potential for JEV at a 1km resolution. JEV spillover potential was estimated by multiplying the ecological suitability index with human population density. We used this estimate to calculate the total population that could be exposed to JEV at the Local Government Area (LGA) level and State/Territory level. We validated our estimates by calculating a population-weighted mean value for each LGA and compared the values between

**Data availability statement:** All data used in this study are openly available from the sources cited in the manuscript. All code used to process the data, construct the indices and generate the results is openly available at: https://github.com/ebskinner/JEV-spillover-risk.

**Funding:** The funders had no role in study design, data collection and analysis, decision to publish, or preparation of the manuscript. This work was supported by the Operational Research and Decision Support for Infectious Diseases (ODeSI) program, which is funded by The University of Queensland's Health Research Accelerator (HERA) initiative (2021–2028). BS was supported by an Australian National Health and Medical Research Council Investigator Grant (GNT2034827). CLL was supported by an Australian National Health and Medical Research Council Investigator Grant (APP1193826). BS, ES, LFK, BJ, GD, CLL were supported by an Australian National Health and Medical Research Council Targeted Call for Research (TCR) grant (2039802). EAM was funded by the US National Institutes of Health (R35GM133439, R01AI168097), the US National Science Foundation (DEB-2011147 with Fogarty International Center), the Stanford Center for Innovation in Global Health, the Stanford Sustainability Accelerator, the Stanford Center for Human-Centered Artificial Intelligence, the Stanford Woods Institute for the Environment, and the Stanford King Center on Global Development.

**Competing interests:** The authors have declared that no competing interests exist.

LGAs with confirmed JEV clinical cases to those without. We found an elevated ecological suitability along the east coast and south-western coast of Australia, inland from the northern centre of the country, and surrounding the Murray River Basin (along the border of New South Wales and Victoria). If JEV becomes established in ecologically suitable areas, high spillover potential to humans would exist along Australia's eastern coast. This exposure potential extends inland to areas like the Murray River Basin, which provide ample habitat for enzootic hosts and vector species. High spillover potential also exists in the Northern Territory, along the southwestern coast of Western Australia, and in South Australia. LGAs with historically confirmed clinical cases in humans had a statistically higher population-weighted mean value compared to those without confirmed cases, supporting the model's capability to differentiate highly suitable areas. By integrating vector-host dynamics and human population density into a spatially explicit framework, we identified areas with high JEV ecological suitability and the potential for spillover into human populations if transmission of JEV were to become established. The results were driven by interactions between vectors, hosts and vector competence. These findings provide insights for targeted surveillance and vector control strategies in Australia. Proactive and sustained interventions are essential to mitigate JEV's growing threat and to protect vulnerable populations in the face of ongoing environmental changes.

## Author summary

Japanese encephalitis virus (JEV) is a mosquito-borne virus that typically circulates between waterbirds and pigs. Although human infections are uncommon, they can cause severe illness or death. Until recently, JEV transmission in Australia was limited to sporadic cases in the tropical north. However, in 2022 and 2023, widespread outbreaks occurred for the first time in temperate southern regions. This raised important questions about whether JEV could become established in new areas of the country. To investigate this risk, we developed a spatial model that maps where environmental conditions are suitable for JEV transmission and where humans might be at highest risk. The model incorporates data on mosquito vectors, animal hosts, and human population density. We found that large parts of eastern and southern Australia, including densely populated regions, may be suitable for JEV spillover to humans. Our findings provide a new tool to guide surveillance, vaccination, and mosquito control, helping protect vulnerable communities as climate and environmental conditions continue to change.

## Introduction

Japanese encephalitis (JE) is a severe central nervous system infection caused by Japanese encephalitis virus (JEV), a mosquito-borne flavivirus transmitted to humans

from waterbird reservoirs and amplifying hosts, including pigs [1]. JEV is the leading cause of viral encephalitis in Asia and is an important cause of epidemic encephalitis in many Southeast Asian and Western Pacific countries [2,3]. The majority of cases occur in children below 15 years of age, but the proportion of cases and outbreaks among adults are increasing in some areas [4]. While acute encephalitis occurs in less than 1% of infections, it often results in death (20–30%) or significant neurological disability (30–50%) when it does occur [5–8]. Unlike most zoonotic and vector-borne diseases, there is a highly effective human vaccine for JEV that can be deployed when there is elevated risk to humans, making the prediction of environmental and climatic conditions critical to inform interventions.

In Australia, JEV is an emerging public health concern. Prior to its designation as a nationally notifiable disease, transmission of JEV had only occurred sporadically in limited areas of tropical northern Australia [9] (Fig 1). In the last four years, however, Australia has experienced a widespread outbreak in southern temperate regions of the country, which reported 46 human cases (including seven deaths) [10,11] and infected at least 80 piggeries between 2022 and 2023 [12–14]. Given that most infections are asymptomatic, it is likely that many more human infections occurred but were not reported. For instance, seroprevalence surveys conducted in rural towns within New South Wales [15] and Victoria [16] found that 8.7% (n = 80) and 3.3% (n = 27) of the participants, respectively, had positive IgG antibodies for JEV, but the majority had never travelled to a JEV endemic country [15]. The large extent and impact of the outbreak, particularly in areas where JEV had not been previously detected or considered a threat, has raised concerns about the virus's potential

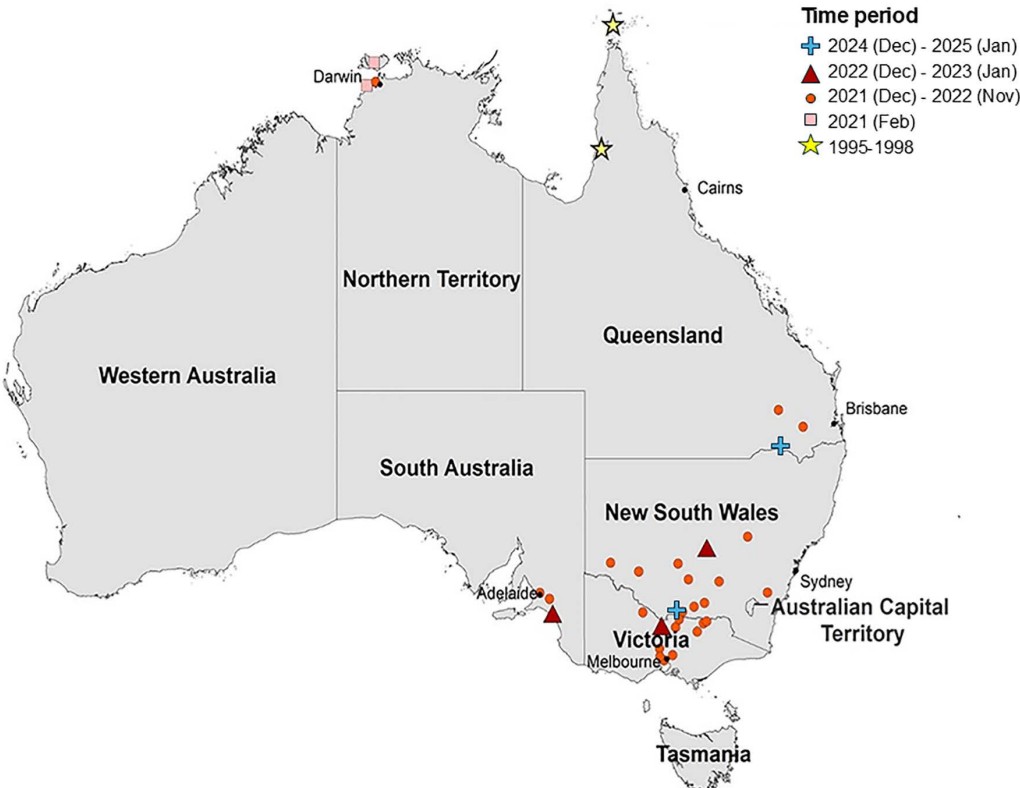

**Fig 1. Map of Australia showing States and Territories discussed in the text, and locations of the human JEV cases in Australia between 2021-2025 (adapted from McGuinness *et al.* 2023 [9], published under a Creative Commons CC BY license).** Base map: Australian state and territory boundaries from the Australian Bureau of Statistics (ABS), Australian Statistical Geography Standard (ASGS), Edition 3 (July 2021–June 2026). Available at: https://www.abs.gov.au/statistics/standards/australian-statistical-geography-standard-asgs-edition-3/jul2021-jun2026. Licensed under Creative Commons Attribution 4.0 International (CC BY 4.0).

to persist and become endemic. In late 2024, JEV was detected in mosquitoes in New South Wales during routine vector surveillance [17], and in sentinel chickens in Western Australia [18]. Since then, there have been two locally acquired confirmed cases of JEV in January 2025 in Victoria and Queensland [19,20], suggesting that endemic transmission may already be established [21]. In March 2025, JEV was detected during mosquito surveillance in Brisbane which is the largest urban centre of Queensland. Prior to this, JEV had been considered a concern in rural areas suggesting potential for urban establishment.

In response to the 2022–2023 JEV outbreak in Australia, the federal government declared a national emergency response and developed a coordinated JEV Outbreak Plan [22]. This response focused on minimising public health risk through vaccination programs, raising awareness among health professionals, and public messaging to avoid mosquito bites as well as enhanced mosquito surveillance and control efforts. During the outbreak more than 125,000 doses of JEV vaccine had been administered [23]. Prior to this, JEV vaccines were only recommended for people living in the outer Torres Strait Islands, some laboratory workers and for people travelling for at least a month in regions where the virus is endemic [15]. While the outbreak response efforts were crucial in addressing immediate public health concerns, they were largely reactive because the transmission dynamics in Australia – particularly the important vectors and hosts involved in JEV transmission – were poorly understood at the time. There was also limited knowledge regarding the geographic areas at risk, the virus's potential to escalate, and how best to target different interventions. The limited understanding made it difficult to predict which areas were at the greatest potential for JEV transmission and spillover.

JEV has a complex transmission ecology, involving multiple mosquito vector species and vertebrate hosts across its geographic distribution. In general, *Culex* spp. mosquitoes are recognised as important vectors, with *Cx. tritaeniorhynchus* being the primary JEV vector in Asia and the Pacific and *Cx. vishnui, Cx. gelidus, Cx. fuscocephala,* and *Cx. pseudovishnui* contributing as secondary vectors [24]. Within Australia, *Cx. annulirostris* is considered to be the most important vector because of its high susceptibility to infection, high competence for transmission, and large geographic distribution across Australia [25–27]. Other studies have suggested that *Cx. gelidus*, *Cx. quinquefasciatus,* and *Cx. sitiens* also have vector potential in Australia based on their competence for transmission and frequently reported feeding on avian hosts [25,26]. Pigs and the Ardeidae family of birds, which include herons and egrets, are strongly implicated as maintenance hosts of JEV in Asia. Within Australia, five of the twelve species of Ardeids have been experimentally infected with JEV and demonstrated high competence, indicating that they could play a role in endemic transmission [28]. Ardeid birds have large distributions across Australia, partially migrate, and have colonial breeding habits which can bring high numbers of birds into close contact with each other and vectors. Pigs are also considered an important potential host for the transmission of JEV due to their high levels of viraemia coupled with their close proximity to humans in farm settings and widespread distribution in feral populations in some parts of rural Australia [28]. Other vertebrate species, such as bats and ducks, may play some role in the transmission of JEV in Australia, however the evidence for this is inconclusive.

This research aims to address gaps in our understanding of JEV transmission by developing mechanistic models that integrate the interactions and competence of vectors and hosts as well as human populations. By synthesising ecological knowledge of JEV transmission pathways, this study seeks to calculate spillover potential driven by vector and host interactions. The specific objectives are to: i) develop predictive maps that identify areas that are ecologically suitable for JEV transmission, combining vector and host ecology with JEV vector competence data; and ii) estimate the spillover potential to human populations for each LGA.

## Results

### Reservoir host distribution sub-models

The predicted probability of reservoir host species occurrence varied substantially across Australia. Ardeidae family distributions were strongly associated with surface water, and, hence, were sparse in the arid interior regions of Australia, specifically large parts of South Australia and Western Australia as well as interior parts of the Northern Territory, Queensland,

and New South Wales (Fig 2a). The distribution of feral pigs ranged from far northern parts of the Northern Territory and from far north Queensland down along the coastline to New South Wales and eastern Victoria, and down to Tasmania (Fig 2b). A small pocket of increased probability of occurrence was also predicted in southwest Western Australia and in northeast Western Australia near the border with the Northern Territory (Fig 2b). Areas with high probability of occurrence were also predicted in New South Wales and Victoria (also around the Murray River Basin), with inland pockets also predicted in southeast Queensland (Fig 2b). The main belt of domestic piggeries stretches from inland southeast Queensland south to Victoria, with a further high concentration in South Australia (Fig 2c).

## Vector sub-models

When considering the vectorial potential for each mosquito species there were distinct spatial differences across Australia. *Cx. annulirostris* demonstrated a high vectorial potential along the eastern coastline and northern regions of the Northern Territory as well as a concentrated high vectorial potential along the Murray River Basin (Fig 3a). There was a moderate vectorial potential for *Cx. annulirostris* extending to inland regions of the eastern coast and throughout Victoria in the south-east corner of the country (Fig 3a). The vectorial potential for *Cx. quinquefasciatus* was similarly concentrated along coastal regions and along the Murray River Basin but showed notable differences in spatial distribution compared to *Cx. annulirostris* (Fig 3b). *Cx. quinquefasciatus* also had a greater vectorial potential in southwestern Australia and in coastal parts of South Australia (Fig 3b). These differences reflect habitat and host availability that is distinct between the two

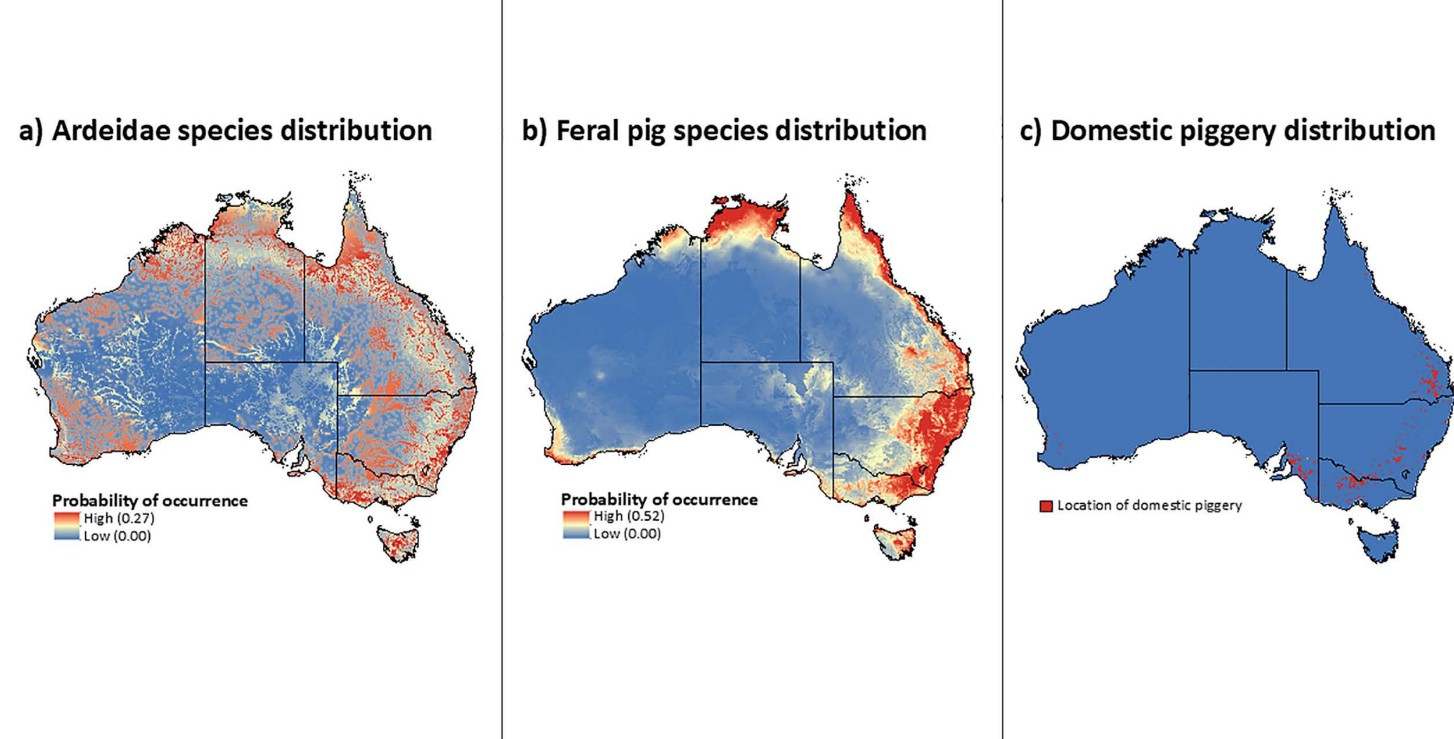

**Fig 2. Predicted host species distributions across Australia with a) Ardeidae species distribution and b) feral pigs distribution calculated from maximum entropy models, and c) domestic piggery distribution obtained from Yakob et al. [29] with 5 km buffer added around each piggery.** Base map: Australian state and territory boundaries from the Australian Bureau of Statistics (ABS), Australian Statistical Geography Standard (ASGS), Edition 3 (July 2021–June 2026). Available at: https://www.abs.gov.au/statistics/standards/australian-statistical-geography-standard-asgs-edition-3/jul2021-jun2026. Licensed under Creative Commons Attribution 4.0 International (CC BY 4.0).

 **PLOS** **Neglected Tropical Diseases**

**a)** *Culex annulirostris* **vectorial potential**

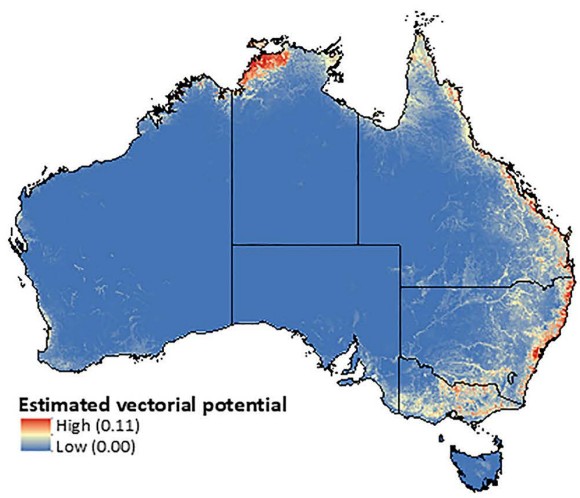

**b)** *Culex quinquefasciatus* **vectorial potential**

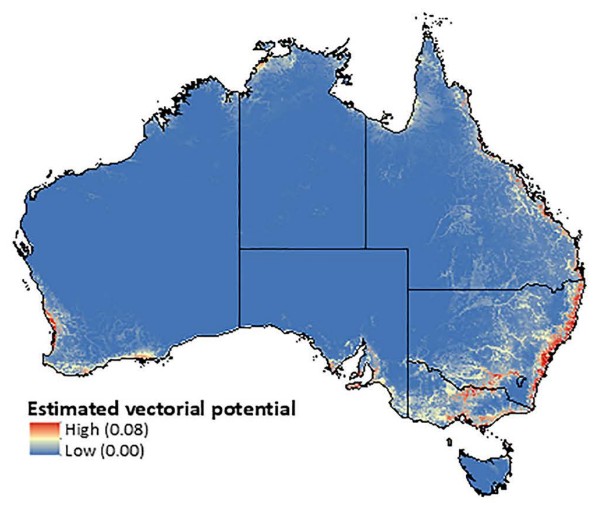

**Fig 3. Vectorial potential for a)** *Cx. annulirostris* **and b)** *Cx. quinquefasciatus* **calculated based on each vector's distribution, overlap with each host, and contact rate for each host, multiplied by the vector's infection and transmission potential for JEV.** Colour scales are species-specific, reflecting the independent calculation of vectorial potential for each mosquito species using species-specific input parameters. Values are therefore not intended for direct comparison in magnitude between panels. Base map: Australian state and territory boundaries from the Australian Bureau of Statistics (ABS), Australian Statistical Geography Standard (ASGS), Edition 3 (July 2021–June 2026). Available at: https://www.abs.gov.au/statistics/standards/australian-statistical-geography-standard-asgs-edition-3/jul2021-jun2026. Licensed under Creative Commons Attribution 4.0 International (CC BY 4.0).

vectors but is also driven by the higher infection and transmission rate for JEV in *Cx. annulirostris* compared to that of *Cx. quinquefasciatus.*

## Ecological suitability index

The ecological suitability index for JEV was built by aggregating the values from the combined contribution of each vector. The ecological suitability index had strong alignment with historic JEV transmission, with predicted high ecological suitability in areas around the Murray River Basin where infections were reported in humans and pigs in the 2022–2023 outbreak (Fig 4, inset). An elevated ecological suitability was also predicted for areas in southwest Western Australia, northern part of the NT, northern QLD, and most of the eastern coastline, and areas with moderate ecological suitability included southwest Victoria, and along waterways throughout western QLD, NSW, and southern WA. Central Australia was not estimated to be ecologically suitable (Fig 4). All ecological suitability index values in our results ranged between 0 and 1. While the model structure does not inherently constrain values to this range, in our implementation each submodel input

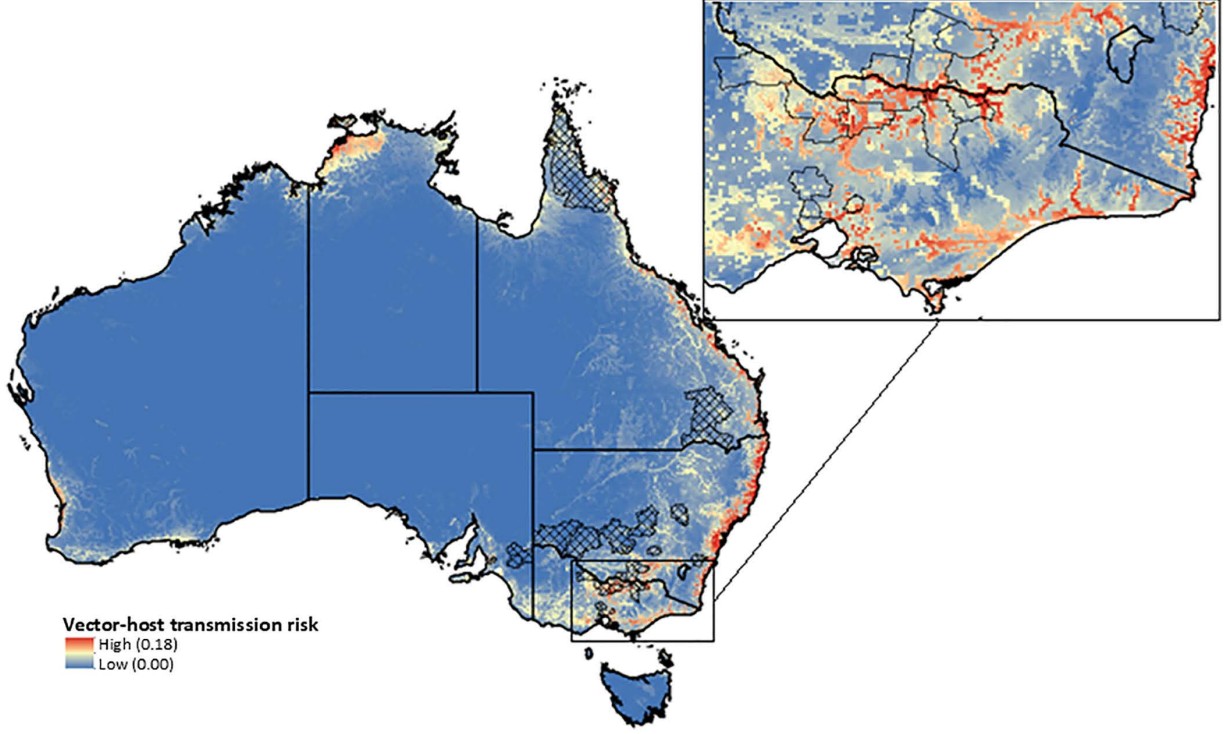

**Fig 4. Ecological suitability index for JEV transmission across Australia (left) with Local Government Areas where historic transmission of JEV has been reported in humans (hashed areas) and around the Murray River Basin in New South Wales/Victoria (LGAs displayed with black border; right).** Base map: Australian state and territory boundaries from the Australian Bureau of Statistics (ABS), Australian Statistical Geography Standard (ASGS), Edition 3 (July 2021–June 2026). Available at: https://www.abs.gov.au/statistics/standards/australian-statistical-geography-standard-asgs-edition-3/jul2021-jun2026. Licensed under Creative Commons Attribution 4.0 International (CC BY 4.0).

was bounded between 0 and 1 and the union formula was applied to combine vector species contributions. This parameterisation, rather than the model framework itself, produced outputs within the 0–1 range.

### Potential for spillover to humans

A total of 2,225,338 people (95% CI: 73,186–9,120,601) or 8.7% of the total population of Australia were estimated to be at risk of potential JEV spillover if transmission were to become established. This was based on the ecological suitability index within each LGA and human population density per 1km$^2$. The highest spillover potential was estimated to be in New South Wales populations with as much as 11.9% of people living in areas with high ecological suitability, followed by Queensland (8.5%) and Victoria (7.9%). Tasmania has the lowest percentage spillover potential, with only 1.0% of its population living in proximity to ecologically suitable areas for JEV transmission (Table 1). The estimated population with JEV spillover potential for each LGA are available in S3 Table. The highest spillover potential is concentrated in areas where there is high human population density in central New South Wales, southeast Queensland, southern Victoria, southwest Western Australia and the Australian Capital Territory (Fig 5). Other capital city areas of Australia, including within South Australia (Adelaide), the Northern Territory (Darwin) and Tasmania (Hobart) had low spillover potential. Within Queensland and New South Wales, there was a moderate spillover potential within LGAs along the east coast of Australia (S9). All remaining populations with some spillover potential were just west of the Great Dividing Range, and in rural Victoria (Fig 7). The only populations estimated with no spillover potential were in the interior of Australia including Mount Isa (Queensland) and City of Kalgoorlie-Boulder (Western Australia).

**Table 1. Potential human population at risk of JEV infection from spillover, overall and by state/territory in Australia (in descending order).**

| State | Total human population | Potential population at risk of spillover infection | Lower 95% confidence interval – population | Upper 95% confidence interval – population | Percentage of population at risk of spillover infection | Lower 95% confidence interval – percentage | Upper 95% confidence interval – percentage |
|---|---|---|---|---|---|---|---|
| National | 25,577,628 | 2,225,338 | 73,186 | 9,120,601 | 8.7% | 0.3% | 35.7% |
| New South Wales | 7,975,199 | 949,679 | 32,730 | 3,774,206 | 11.9% | 0.4% | 47.3% |
| Queensland | 5,223,770 | 443,626 | 17,239 | 1,846,559 | 8.5% | 0.3% | 35.3% |
| Victoria | 6,592,291 | 523,218 | 16,698 | 2,183,503 | 7.9% | 0.3% | 33.1% |
| Western Australia | 2,742,961 | 187,586 | 3,623 | 790,708 | 6.8% | 0.1% | 28.8% |
| Australian Capital Territory | 456,522 | 23,861 | 873 | 100,484 | 5.2% | 0.2% | 22.0% |
| Northern Territory | 244,266 | 11,765 | 483 | 51,927 | 4.8% | 0.2% | 21.3% |
| South Australia | 1,799,428 | 80,314 | 1,454 | 349,211 | 4.5% | 0.1% | 19.4% |
| Other Territories | 4,788 | 57 | 2 | 220 | 1.2% | 0.0% | 4.6% |
| Tasmania | 538,404 | 5,234 | 82 | 23,782 | 1.0% | 0.0% | 4.4% |

## Validation

LGAs with historically confirmed clinical cases had a statistically higher population-weighted mean value (p < 0.001) compared to those without confirmed cases, supporting the model's ability to differentiate spillover potential in different areas (S5 Fig). The population-weighted mean was calculated for each LGA by multiplying the ecological suitability index by human population density at the 1 km grid level, then averaging these values weighted by the population within the LGA. In addition, the visual comparison showed strong alignment between the model's predicted areas of elevated spillover potential and LGAs with reported historic JEV cases.

## Discussion

The 2022–2023 JEV outbreak in Australia and human deaths in 2025, highlighted the lack of knowledge on the ecological conditions that lead to transmission. This is a crucial first step to better identify areas that are suitable for JEV transmission and which human populations may have potential spillover if JEV becomes locally established in suitable environments. Previous studies have explored various environmental factors influencing JEV transmission in Australia, such as landscape features linked to outbreaks [30], the overlap of vector and host or human distributions [29,31], and meteorological drivers of transmission [32]. Our study advances this field by integrating multiple ecological processes (including vector-host distributions, transmission probabilities, and human population density) into a mechanistic model to understand the ecological suitability and the potential for spillover into human populations. Our study identifies areas with ecological suitability for JEV in New South Wales and Victoria (particular along the Murray River Basin bordering these two states) and Queensland, highlighting the need for targeted surveillance and intervention strategies.

The ecological suitability index capture processes essential for disease transmission. These processes, such as the presence of competent vectors, suitable hosts and favourable environmental conditions, are each necessary components for transmission but cannot on their own sustain outbreaks. Only when these processes align can local transmission occur and lead to spillover to humans [33,34]. In other multi-vector, multi-host disease systems, integrating ecological and human population drivers has been critical for predicting transmission [35]. For instance, in predicting yellow fever virus

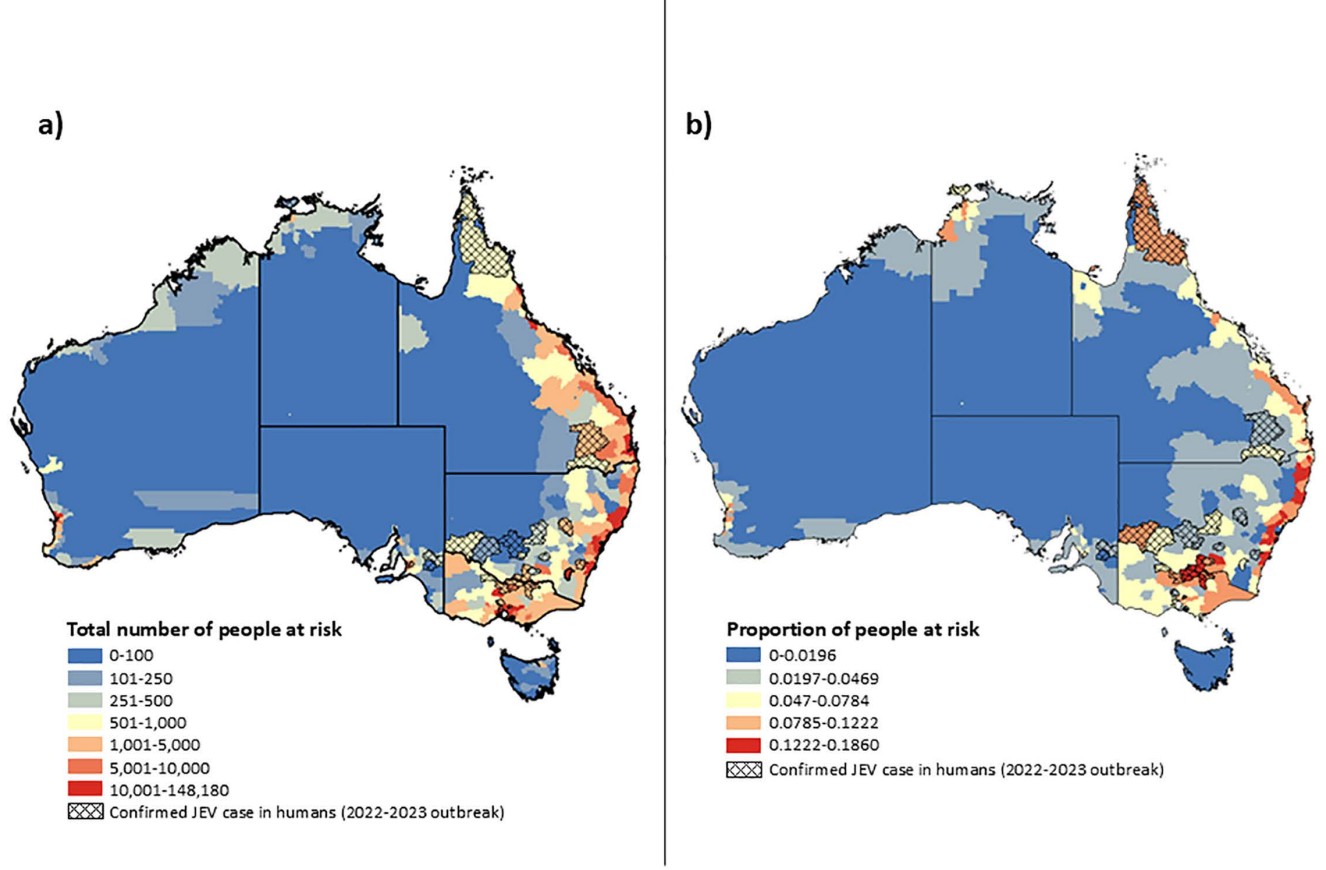

**Fig 5. Estimated a) total population and b) proportion of population with JEV spillover potential within each Local Government Area (LGA).** Hashed LGAs indicate areas with historically confirmed human cases. Base map: Australian state, territory, and LGA boundaries from the Australian Bureau of Statistics (ABS), Australian Statistical Geography Standard (ASGS), Edition 3 (July 2021–June 2026). Available at: https://www.abs.gov.au/statistics/standards/australian-statistical-geography-standard-asgs-edition-3/jul2021-jun2026. Licensed under Creative Commons Attribution 4.0 International (CC BY 4.0).

transmission in Brazil, ecological suitability was a highly accurate predictor of observed spillover in people, while metrics incorporating human population density and vaccination alone were less accurate [36]. Here, our ecological suitability index identified elevated suitability in most (but not all) LGAs where JEV transmission has historically occurred in Australia. This indicates that while the model can capture broad patterns of spillover potential, some areas of past outbreaks were not identified as highly suitable, likely reflecting limitations in the available ecological and demographic data or unmeasured local factors that may have influenced transmission. However, the model also identified elevated suitability in areas where locally acquired JEV transmission has not been reported. These false positives are expected because the model will identify all areas of potential suitability based on the ecological suitability. This could imply one or more things about these regions: they are environmentally suitable, but an outbreak has not yet occurred (but could in the future); they are environmentally suitable, and outbreaks have gone undetected; or they are not environmentally suitable, and the model is overpredicting habitat suitability by missing important constraints. Distinguishing between these scenarios would require additional data, such as enhanced surveillance for undetected circulation, field assessments of vector and host abundance and exposure, and localised environmental data to better understand the conditions that might influence vector-host interactions and disease transmission. We hypothesise that the regions that have high ecological suitability

represent areas where JEV transmission could occur under favourable conditions (i.e., model outputs identify potential and not actual transmission) given the availability of interacting competent vectors and hosts.

In comparison to another study on the enzootic and epidemic transmission risks of JEV in Australia, our ecological suitability index estimates offer a more refined geographic profile, predicting a reduced area where transmission could feasibly occur [31]. This improvement is due to our mechanistic model, which integrates not only the probability of host and vector occurrence but also the unique competence of each vector and scaled interactions between vectors and hosts. By capturing these additional ecological and transmission factors, our model provides a best-evidence, multi-factor assessment that enhances predictions of where the environment is suitable for JEV spillover.

Across Australia, we estimated a total of ~2.2 million people at risk of JEV from spillover, representing approximately 8.7% of the Australian population. This is a higher estimate than a previous study which estimated 740,546 people could potentially be at risk of JEV exposure [29] (equating to approximately 3% of the Australian population), which was calculated based on the number of people within 4.4km (the maximum flight range of *Cx. annulirostris*) of a piggery. Our estimate accounts for multiple pathways of transmission, including the interactions between different vectors and hosts, which broadens the areas considered suitable. In our model, we considered the distributions and abundance of both domestic and feral pigs and birds as potential hosts. While this approach assumes that pigs and birds contribute proportionally to their abundance and competence as reservoirs, the biological reality is much more complex. For example, in Asia where JEV is endemic, pigs are considered to be amplification hosts [37], playing an important role in increasing viral load in some contexts, while Ardeidae species are considered important maintenance or reservoir hosts. These distinctions in host roles may influence spillover dynamics to human populations, but further research is needed to quantify their relative contributions under Australian ecological conditions. Further, the role of domestic piggeries as a source of human infections in Australia has not been confirmed [28,38]. Within piggeries, workers may be more likely to be vaccinated against JEV and pigs with confirmed infection removed, which collectively can limit the spillover potential to humans. This broader assumptions of potential transmission pathways may explain why our estimate is more than double what was previously reported based on domestic piggeries alone.

By applying distinct measures of spillover risk—ecological suitability and human population in proximity—we capture distinct aspects of the transmission process, which in turn can inform different management applications. For example, our ecological suitability index, available at a $1km^2$ resolution across Australia, identifies areas where JEV transmission between species is high, enabling targeted ecological interventions. These may include strategies such as reducing vector populations through habitat modification, insecticide applications or biological control, as well as implementing enhanced vector and host surveillance to detect early signs of virus transmission. Such interventions aim to disrupt transmission at its source, preventing the virus from spilling over into human populations. In contrast, our human spillover potential model, aggregated to LGAs could be used for guiding public health responses. These include assessing resource allocation for healthcare services and designing public messaging campaigns to encourage behaviours that reduce mosquito exposure. By using both models in tandem, we can address the complexity of JEV transmission dynamics and tailor interventions to specific stages of transmission which can enhance Australia's preparedness to respond proactively to JEV cases.

Our findings do not consider the potential seasonality of JEV or the broader implications of temperature or rainfall patterns on vector abundance and competence. Both environmental and human spillover estimates are static, yet they manage to capture patterns of where previous JEV transmission has occurred in Australia by incorporating mechanistic understanding of JEV transmission. Incorporating temporal scales in future studies (e.g., coupling the framework with seasonal climate forecasts, time-series analysis, or mechanistic models of vector population dynamics) would allow identification of seasonal hotspots, facilitating more dynamic assessments. Climate is an important moderator of JEV transmission as it has the potential to shift seasonal host distributions, vector abundance, vector competence, and vector life history traits that drive transmission. Presently, the effects of climate (or of a changing environment) on JEV transmission are poorly understood and require more empirical data and research such as vector competence experiments across

temperatures for multiple hosts and an understanding of how feeding patterns and vector and host distributions may shift. Given the increasing urgency of understanding drivers of JEV transmission across Australia, this study aims to synthesise ecological and human population information on key drivers of the spillover process from somewhat limited empirical data in Australia. Vector and host occurrence data, along with bloodmeal data —which form the basis of the probability of occurrence models used—are likely biased towards relatively more densely populated areas. Further, while the vector distribution maps used were drawn from published sources, they are based on a low number of reported occurrences in Australia, potentially limiting their accuracy and ability to pick up on key environmental limitations. Another limitation of our study is the use of a family-level ecological niche model rather than species-specific models for Ardeidae, which may overlook the distinct ecological preferences of individual species within the Ardeidae family. We will address this issue in our future work by developing species-level models. This is crucial as we know almost nothing about which birds are important in Australia, although Ardeidae throughout Asia seem highly competent. While evidence is currently inconclusive, other vertebrate hosts [39,40] could potentially contribute to JEV transmission in Australia and were thus not accounted for in our present estimates. As more evidence emerges, we will incorporate these data for these other potential vertebrate hosts in future iterations of this work. Despite the data limitations, the modelled species distributions align strongly with entomological and ecological understandings for each vector species. The strong alignment between our mechanistic model predictions and observed outbreaks suggests that despite the limitations of the available data, a mechanistic understanding of the ecological suitability can provide useful predictions of spillover potential, with applications for public health.

## Conclusions

The unprecedented outbreak of JEV between 2022 and 2023 as well as the recently confirmed human cases in 2025 pose significant public health challenges in Australia, exposing critical gaps in our understanding of the disease's transmission dynamics. This study addresses these gaps by integrating the distributions and interactions of vectors and hosts, vector competence, and human population density to develop metrics. JE is a deadly disease that represents a growing threat to people and domestic animals in Australia and beyond. Understanding the drivers of JEV transmission is crucial for guiding disease control efforts and public health measures. Specifically, when and where JEV potential is elevated, practitioners across public health, vector control, agrobusiness, wildlife conservation, government, and academic sectors can implement a range of strategies to limit human and animal transmission, which may include public health messaging, expanded human, wildlife, and domestic animal surveillance, targeted vector control, quarantine, and more.

The ecological suitability index and human spillover potential developed in this study provide a foundation for future work and can be refined as additional empirical data become available. Improving the predictive power will require integrating temporal variability, climate and other environmental changes affecting reservoir host abundance and distribution, as well as shifts in human–animal contact patterns. Such integration would provide a more dynamic understanding of JEV transmission potential in Australia and support timely, targeted interventions.

## Methods

### Conceptual framework

Building on recent advances in modelling pathogen spillover potential as a function of environmental and human factors, we apply a spatially and temporally explicit modelling framework developed in Plowright et al. [34] and applied to yellow fever virus in Brazil by Childs et al. [36]. This approach works by modelling the ecological processes expected to drive transmission within vectors and hosts (i.e., vector distribution, host distribution, vector-host contact, vector competence), and the potential spillover to humans (i.e., human population density and distribution) (Fig 6). Given the very low vaccine coverage in Australia, with programs limited to high-risk areas, and the lack of publicly available data, vaccination rate was not included in the current estimates, and its limited uptake is unlikely to substantially affect transmission dynamics. We then use these individual data streams and submodels to develop two metrics at high spatial resolution (1x1km): i)

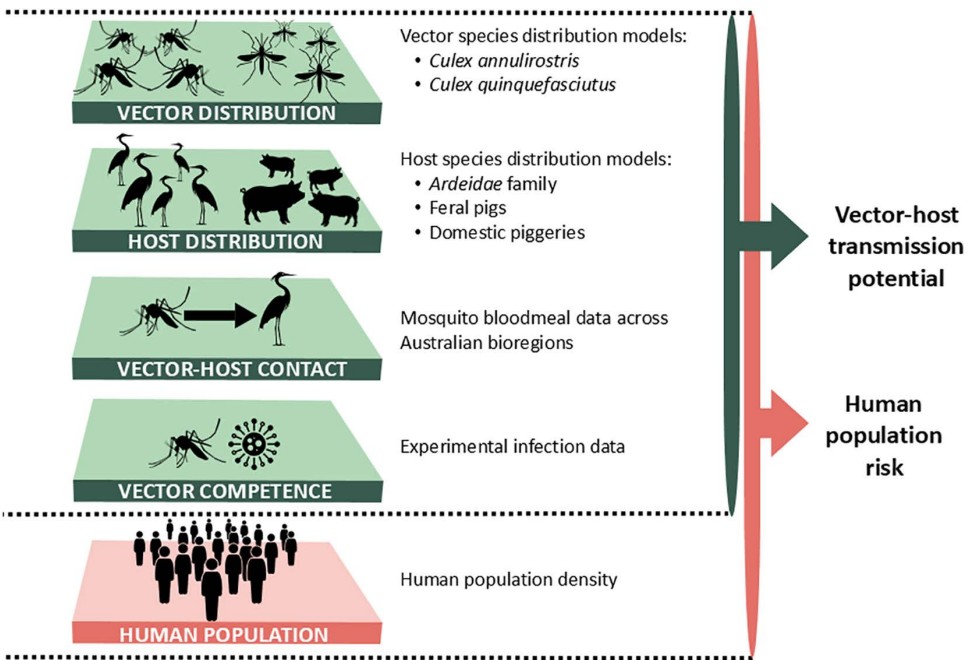

**Fig 6. Conceptual framework for estimating ecological suitability index and human spillover potential for JEV in Australia.**

ecological suitability index for transmission as a function of the vector and reservoir host ecology, and ii) human spillover potential, which estimates the number of people that could potentially be exposed if transmission becomes established. Each ecological or human population factor was parameterised into a submodel, where we used published data to define the relationships and input variables specific to vector and host distributions, vector competence, and human exposure potential (building on the approach in Childs et al. [36]). This research complies with the GATHER statement (S1 Table) [41]. All data used in this study are openly available from the sources cited in the manuscript. All code for data processing, analysis, and figure generation is openly available at https://github.com/ebskinner/JEV-spillover-risk.

Our analysis included two species of mosquito vectors (*Cx. annulirostris* and *Cx. quinquefasciatus*) which occur endemically in Australia, represent distinct ecological traits and habitat preferences (including freshwater and peri-urban species), and have experimentally demonstrated competence for JEV transmission. *Culex sitiens* was originally considered for inclusion in the analysis but had insufficient documented occurrence points (<100 individual mosquitoes) to be confident in the species contribution to the model. For potential hosts in Australia, we included data on the Ardeidae family of birds and on pigs (both domestic *Sus scrofa domesticus* and feral *Sus scrofa*). The potential contribution of other vertebrate hosts for JEV transmission was not considered due to current inconclusive evidence and lack of experimental infection data and virus detections in free-living populations.

Ecological suitability index (or combined vectorial potential probabilities to realise JEV transmission) was calculated at a 1 km² resolution as follows. First, the transmission potential for each of the two mosquito vector species (*Cx. annulirostris* and *Cx. quinquefasciatus*) was calculated using the notion shown in Fig 7, where for example $V_1A = PP_{V1} \times (PP_A \times BP_{V1,A}) \times IP_{V1} \times TP_{V1}$ represents the vectorial potential for *Cx. annulirostris* with Ardeidae hosts. Here, $PP$ denotes the probability of occurrence (sub-models described in next section), $BP$ the probability of a bloodmeal from the given host, $IP$ the infection probability, and $TP$ the transmission probability. Uncertainty ranges for these parameters (S2 Table) were incorporated into the estimation process to provide 95% uncertainty intervals. Second, for each vector species, the total vectorial potential was calculated as the sum of the contributions from Ardeidae birds, feral pigs, and domestic pigs (e.g.,

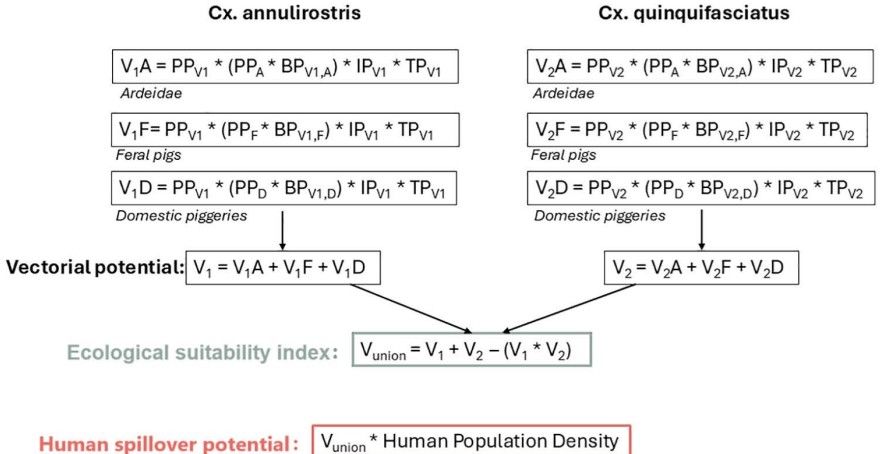

**Species:** Vector (V); Ardeidae (A), Feral pigs (F), Domestic Pigs (D)
**Constants:** Presence probability (PP); Bloodfeeding Probability (BP); Infection probability (IP); Transmission probability (TP)

**Cx. annulirostris**

$V_1A = PP_{V1} * (PP_A * BP_{V1,A}) * IP_{V1} * TP_{V1}$
*Ardeidae*

$V_1F = PP_{V1} * (PP_F * BP_{V1,F}) * IP_{V1} * TP_{V1}$
*Feral pigs*

$V_1D = PP_{V1} * (PP_D * BP_{V1,D}) * IP_{V1} * TP_{V1}$
*Domestic piggeries*

**Cx. quinquifasciatus**

$V_2A = PP_{V2} * (PP_A * BP_{V2,A}) * IP_{V2} * TP_{V2}$
*Ardeidae*

$V_2F = PP_{V2} * (PP_F * BP_{V2,F}) * IP_{V2} * TP_{V2}$
*Feral pigs*

$V_2D = PP_{V2} * (PP_D * BP_{V2,D}) * IP_{V2} * TP_{V2}$
*Domestic piggeries*

**Vectorial potential:** $V_1 = V_1A + V_1F + V_1D$    $V_2 = V_2A + V_2F + V_2D$

**Ecological suitability index:** $V_{union} = V_1 + V_2 - (V_1 * V_2)$

**Human spillover potential:** $V_{union} *$ Human Population Density

**Fig 7. Modelling framework used to calculate the ecological suitability index and human spillover potential for Japanese encephalitis virus in Australia, linking presence probability (or probability of occurrence) for main proposed vectors and hosts, including adjustments for infection potential, transmission potential, and blood host preference and, for human spillover potential, multiplied by human population density.**

$V_1 = V_{1A} + V_{1F} + V_{1D}$ in Fig 7). This summation gives equal weighting to each host group because empirical data on the relative contribution of these hosts to JEV amplification in Australia are lacking. Equal weighting was therefore adopted as a pragmatic assumption, and can be refined when host-specific contribution estimates become available. Finally, the ecological suitability index (Fig 7: $V_{union}$) was calculated by combining the two vector species as $V_{union} = V_1 + V_2 - (V_1 \times V_2)$. This formulation assumes statistical independence between the two vector species in terms of their spatial occurrence and potential for JEV transmission. This assumption is reasonable in the Australian ecological context, where *Cx. annulirostris* and *Cx. quinquefasciatus* occupy distinct but sometimes overlapping ecological niches (freshwater versus peri-urban habitats), and co-occurrence at fine spatial scales is limited.

All component probabilities used in calculating vectorial probability potential for JEV transmission (namely, the probability of vector presence (PP), bloodmeal preference (BP), infection probability (IP), and transmission probability (TP)) are theoretically bounded in probability space, i.e., 0 and 1. These were combined multiplicatively for each host-vector interaction, under the simplifying assumption of independence between parameters due to the lack of empirical data on their joint distributions. For each mosquito species (*Cx. annulirostris* and *Cx. quinquefasciatus*), the total vectorial potential (Vi) was calculated as the sum of contributions from three host groups (Ardeidae birds, feral pigs, and domestic pigs). While the theoretical maximum for each $V_i$ could approach 1 under perfect conditions (i.e., all input probabilities equal to 1), the combined ecological suitability index ($V_{union}$), calculated using standard probability union formula, also has a theoretical upper bound just below 1. In ecological terms, values near 0 indicate minimal suitability for JEV transmission, reflecting low probability of vector or host presence, low contact rates, or low vector competence. Values approaching 1 represent the highest suitability, where competent vectors and amplifying hosts co-occur with maximum probability, contact, and transmission potential. However, in practice, the empirically derived input parameters—based on literature estimates for contact rates, vector competence, and occurrence probabilities—are substantially lower than their theoretical maxima. As a result, the observed ecological suitability index values in our model ranged from 0 to 0.18, indicating that the maximum vectorial potential for JEV transmission under current ecological conditions is approximately 18%.

Human spillover potential was then calculated per 1 km² by multiplying the ecological suitability index ($V_{union}$) with human population density (Fig 7). This index has no fixed upper bound because it scales with population density; higher

**PLOS** **Neglected Tropical Diseases**

values represent areas where ecological suitability coincides with large human populations, thereby indicating greater potential for human exposure should transmission occur. We then aggregated this spillover potential by Local Government Area (LGA) (a population-based administrative jurisdiction, third-level government division), using an LGA boundaries shapefile in ArcGIS, to estimate spillover potential in human populations per LGA and State.

## Sub-models

**Vectors.** Raster files representing the probability of vector presence for *Cx. annulirostris* and *Cx. quinquefasciatus* were obtained from the ecological niche models developed by Furlong et al. [24], with layers are available at: https://doi.org/10.25919/r017-5816. Infection and transmission rates for each *Culex* species were extracted from experimental studies undertaken within Australia (synthesised by van den Hurk *et. al.* 2022 [23]). We extracted the minimum, average, and maximum reported infection and transmission values between studies to generate an upper and lower model uncertainty estimate [25] (S2 Table). Vector-host contact rates were derived from a meta-analysis conducted in 2019 that compiled all known data on bloodmeals from target mosquito species in Australia, which included 6,089 bloodmeals for *Cx. annulirostris* and 4,746 bloodmeal records for *Cx. quinquefasciatus* [42] (S2 Table). To generate upper and lower estimates and account for the variation of mosquito feeding patterns across space and time, we aggregated bloodmeal studies by the climate region where the data was collected (i.e., equatorial, tropical, subtropical, and temperate) and calculated the minimum, average, and maximum proportion of bloodmeals from birds and pigs between climate regions.

**Hosts.** We applied an ecological niche model using a maximum entropy approach to predict probability of occurrence for species in the Ardeidae family in Australia by using observed occurrence data (i.e., presence-only estimates) from the Atlas of Living Australia for the period 2013–2023 (S1 Fig). Climate and environmental variables were sourced from the Australia Bureau of Meteorology (http://www.bom.gov.au/climate/) and Geoscience Australia (https://www.ga.gov.au/). The final covariates used in the maximum entropy model for this species were distance to nearest surface water, rainfall, and evapotranspiration.

For feral pigs (*Sus scrofa*), we created an ecological niche model using a maximum entropy approach to predict probability of occurrence using observed occurrence data (i.e., presence-only estimates) from the Australian Atlas of Living Australia for the period 2018–2023 (S2 Fig). The final covariates used in the feral pig maximum entropy model were Normalised Difference Vegetation Index (NDVI) (average for October 2018 to March 2019, latest available grid estimates from BoM, http://www.bom.gov.au/) and rainfall.

The coordinates of domestic piggeries in Australia were obtained from publicly available data provided by the Farm Transparency Project (https://farmtransparency.org). Locations of JEV infected piggeries in 2022 were utilised from Yakob et al [29] (S3 Fig). We create 5km buffer around each piggery to reflect the mostly likely maximum travel distance for mosquito vectors and allow comparison to previous research on JEV transmission risk in Australia [43] and the geographic range of exposure linked to each piggery for our estimations.

**Human population density.** Human population density estimates were obtained from the Australian population grid for 2022 (S4 Fig). The grid is a modelled 1 km x 1 km grid representation of the estimated resident population of Australia from 30 June 2022. The population grid is created by reaggregating estimated resident population data from Statistical Areas Level 1 (SA1) to a 1 km x 1 km grid across Australia based on point data representing residential address points, where each grid represents the number of people per square kilometre. We multiplied the ecological suitability index with the human population density at 1x1km resolution to estimate the human spillover potential. We then aggregated this count to estimate the spillover transmission potential in each LGA and at the State level.

## Mapping

Vector species potential, ecological suitability index and human spillover potential was modelled and estimated using Google Earth Engine (code available at: https://github.com/ebskinner/JEV-spillover-risk). Mapping and visualisation of submodels were performed in ArcMap 10.8.2.

## Validation

Given the low reported number of human cases and that JEV is an emerging infectious disease in Australia with only a few events we validated the model broadly in the following way; we first calculated a population-weighted mean value for each LGA and compared the values between LGAs with confirmed clinical cases and those without. This allowed us to assess whether LGAs with cases were associated with higher modelled spillover potential. This approach provided a quantitative basis for evaluating the model predictions while accounting for the expected high number of false positives. We also visually compared the areas of elevated spillover potential identified by the model to LGAs that reported clinical cases historically. This qualitative comparison was conducted to further assess whether the model aligned well with observed patterns of spillover potential, despite the inherent challenges in quantifying performance with sparse outbreak data.

## Supporting information

**S1 Table. Guidelines for Accurate and Transparent Health Estimates Reporting: the GATHER statement.**
(DOCX)

**S2 Table. Parameter assumptions used in the modelling based on published studies.**
(DOCX)

**S1 Fig. Infection risk probability for water bird (*Ardeidae*).** Small map panel displays observed occurrence data from 2013–2023 (Atlas of Living Australia) used in the maximum entropy model component.
(TIF)

**S2 Fig. Infection risk probability for feral pigs (*Sus scrofa*).** Small panel displays observed occurrence from 2018–2023 (Atlas of Living Australia) used in the maximum entropy model.
(TIF)

**S3 Fig. Locations of piggeries, with those in red having had JEV-positive pigs during the recent outbreaks (Farm Transparency Project) [ref 29].**
(TIF)

**S4 Fig. Human population density estimates in Australia (2022) from the Australian Population Grid 2022 (ABS).**
(TIF)

**S3 Table. Potential human population at risk of JEV infection from spillover by LGA with count and percent at risk, including 95% uncertainty interval (UIs).**
(DOCX)

**S5 Fig. Validation of population at risk by LGA and previously reported JEV cases [ref 9] by LGA (\*Wilcoxon rank-sum (Mann–Whitney) test).**
(TIF)

## Acknowledgments

We would like to thank Dr Tanya Russell, Dr Tatiana Proboste Ibertti and Dr Megan Young for their valuable input and feedback regarding this work. We would also like to thank Dr Zhi Chen for assisting with the interactive webpage development.

## Author contributions

**Conceptualization:** Eloise B. Skinner, Benn Sartorius, Luis Furuya-Kanamori, Adam T. Craig, Behzad Kiani, Brian J. Johnson, Roslyn I. Hickson, Erin A. Mordecai, Gregor Devine, Colleen L. Lau.

**Data curation:** Eloise B. Skinner, Benn Sartorius.

**Formal analysis:** Eloise B. Skinner, Benn Sartorius.

**Funding acquisition:** Benn Sartorius.

**Investigation:** Eloise B. Skinner, Benn Sartorius, Adam T. Craig, Behzad Kiani, Brian J. Johnson, Kevin T. Moore, Roslyn I. Hickson, Erin A. Mordecai, Gregor Devine, Colleen L. Lau.

**Methodology:** Eloise B. Skinner, Benn Sartorius, Luis Furuya-Kanamori, Roslyn I. Hickson, Erin A. Mordecai, Gregor Devine, Colleen L. Lau.

**Project administration:** Eloise B. Skinner, Benn Sartorius.

**Resources:** Adam T. Craig, Behzad Kiani, Colleen L. Lau.

**Software:** Eloise B. Skinner.

**Supervision:** Colleen L. Lau.

**Validation:** Benn Sartorius, Brian J. Johnson, Erin A. Mordecai, Gregor Devine, Colleen L. Lau.

**Visualization:** Eloise B. Skinner, Colleen L. Lau.

**Writing – original draft:** Eloise B. Skinner, Benn Sartorius, Colleen L. Lau.

**Writing – review & editing:** Eloise B. Skinner, Benn Sartorius, Luis Furuya-Kanamori, Adam T. Craig, Behzad Kiani, Brian J. Johnson, Kevin T. Moore, Roslyn I. Hickson, Erin A. Mordecai, Gregor Devine, Colleen L. Lau.

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
