## [Decision Letter · Decision Letter 0]

22 Jun 2025

Ecological suitability of Japanese encephalitis virus in Australia: a modelling analysis of vector-host transmission dynamics to potential spillover in humans

Dear Dr. Sartorius,

Thank you for submitting your manuscript to PLOS Neglected Tropical Diseases. After careful consideration, we feel that it has merit but does not fully meet PLOS Neglected Tropical Diseases's publication criteria as it currently stands. Therefore, we invite you to submit a revised version of the manuscript that addresses the points raised during the review process.

Please submit your revised manuscript within 60 days Aug 21 2025 11:59PM. If you will need more time than this to complete your revisions, please reply to this message or contact the journal office at plosntds@plos.org. Please include the following items when submitting your revised manuscript:

We look forward to receiving your revised manuscript.

Kind regards,

Maira Aguiar, PhD

Academic Editor

Qu Cheng

Section Editor

Shaden Kamhawi

co-Editor-in-Chief

Paul Brindley

co-Editor-in-Chief

**Additional Editor Comments (if provided):**

Please revise your manuscript, taking into consideration the reviewers' suggestions. Ensure that all comments are addressed thoroughly and that any changes are clearly indicated in your revised submission.

**Journal Requirements:**

4) We notice that your supplementary Figures, and Tables are included in the manuscript file. Please remove them and upload them with the file type 'Supporting Information'. Please ensure that each Supporting Information file has a legend listed in the manuscript after the references list.

Potential Copyright Issues:

i) Figure 1 appears to have been previously published. Please provide written permission from the copyright holder to publish this under our CC-BY 4.0 license, or remove the figure / replace the image. Please note we do not recommend using standard request forms available on Publishers' websites, as they grant single use rather than republication under an open access license.

ii) Figures 1, 2, 3, 4, 5, and S3-S6. Please (a) provide a direct link to the base layer of the map (i.e., the country or region border shape) and ensure this is also included in the figure legend; and (b) provide a link to the terms of use / license information for the base layer image or shapefile. We cannot publish proprietary or copyrighted maps (e.g. Google Maps, Mapquest) and the terms of use for your map base layer must be compatible with our CC BY 4.0 license.

6) We note that your Data Availability Statement is currently as follows: "All relevant data used in this study are clearly referenced with source links to where these data can be accessed.". Please confirm at this time whether or not your submission contains all raw data required to replicate the results of your study. Authors must share the “minimal data set” for their submission. PLOS defines the minimal data set to consist of the data required to replicate all study findings reported in the article, as well as related metadata and methods (https://journals.plos.org/plosone/s/data-availability#loc-minimal-data-set-definition).

7) Please amend your detailed Financial Disclosure statement. This is published with the article. It must therefore be completed in full sentences and contain the exact wording you wish to be published.

2) If any authors received a salary from any of your funders, please state which authors and which funders..

8) Please send a completed 'Competing Interests' statement, including any COIs declared by your co-authors. If you have no competing interests to declare, please state "The authors have declared that no competing interests exist". Otherwise please declare all competing interests beginning with the statement "I have read the journal's policy and the authors of this manuscript have the following competing interests".

**Reviewers' Comments:**

Reviewer's Responses to Questions

**Key Review Criteria Required for Acceptance?**

**Methods**

-Are the objectives of the study clearly articulated with a clear testable hypothesis stated?

-Is the study design appropriate to address the stated objectives?

-Is the population clearly described and appropriate for the hypothesis being tested?

-Is the sample size sufficient to ensure adequate power to address the hypothesis being tested?

-Were correct statistical analysis used to support conclusions?

-Are there concerns about ethical or regulatory requirements being met?

Reviewer #1: The objectives of the study are clearly articulated, focusing on the development of a spatially explicit modelling framework to estimate ecological suitability for Japanese encephalitis virus (JEV) transmission and potential human spillover across Australia. Although a formal hypothesis is not explicitly stated in a traditional sense, the goals are specific, testable, and well defined. The study design is appropriate for addressing these objectives, employing a mechanistic model that integrates ecological, entomological, and human demographic data in a spatial context. The populations under study—specifically the vectors (Cx. annulirostris and Cx. quinquefasciatus), hosts (Ardeidae birds, domestic and feral pigs), and human populations—are well described and relevant for the aims of the analysis.

The sample size, in terms of ecological and occurrence data points, is appropriate for the modelling approach and provides sufficient geographic and taxonomic coverage to ensure robustness. The statistical methods used are appropriate and well justified, including maximum entropy modelling, integration of vector-host contact and competence parameters, and population-weighted validation techniques. These methods adequately support the study's conclusions. No concerns were identified regarding ethical or regulatory compliance, as the study relies solely on publicly available and non-identifiable data, with a clear statement included regarding the role of funding sources and data transparency.

Reviewer #2: - In the construction of any index, it is essential to explicitly state the possible range of values the index can assume, as well as the interpretation of those values. The authors should clarify these aspects for both the ecological suitability index and the human spillover potential index.

- Figure 7 presents the structure used to construct the indices. It is recommended that the main text adopt the same notation as the equations and explicitly demonstrate how the probabilities shown in Figure 7 are derived. While the relevant information is provided in subsections, a direct connection with the formal notation is crucial for clarity and reproducibility.

- In Figure 7, the formulas for each species (VA, VF, and VD) are calculated as the product of probabilities. This implies an assumption of independence between the events of transmission, infection, presence, and bloodfeeding. Is this assumption reasonable? If so, it should be explicitly stated in the text. Moreover, as these components are products of probabilities between 0 and 1, each individual term also ranges from 0 to 1. Consequently, the sum of VA, VF, and VD could range from 0 to 3. Why was equal weighting assigned to each of these components? Do the three features considered (Ardeidae birds, feral pigs, and domestic pigs) indeed contribute equally to vector potential? Additionally, the vector potentials calculated for Culex annulirostris and Culex quinquefasciatus are treated as independent. Is this assumption valid in the ecological context of Australia? If so, this should also be justified in the manuscript. If these interpretations are correct, then the ecological suitability index could range from –3 to 3. In that case, what do high values (either positive or negative) mean? And what about values close to zero? The same line of reasoning applies to the human spillover potential index. If these interpretations are not accurate, the authors should rigorously clarify the actual properties of the presented indicators.

- The interpretation of intermediate values leading to the final indices should be discussed in conjunction with the results shown in Figures 3 and 4. For example, subfigures A) and B) of Figure 3 are not directly comparable, as their color scales have different maximum values: the dark red in A) does not represent the same intensity as in B). This difference should be standardized or justified to allow meaningful comparisons.

- In the legend of Figure 4, the term “vector-host transmission risk” is used. However, the map actually displays the ecological suitability index. It is important to clarify that this index does not directly measure transmission risk, nor is a formal definition of “risk” provided in the manuscript. A similar issue occurs in Table 1. Furthermore, although the values shown on the maps range from 0 to 1, this is not necessarily guaranteed by the model described in Figure 7, and this consistency needs to be revised or justified.

- Finally, the authors should state which software was used for data processing and index construction. It is strongly recommended that all data and code used in the analysis be made publicly available in a suitable repository.

**Results**

-Does the analysis presented match the analysis plan?

-Are the results clearly and completely presented?

-Are the figures (Tables, Images) of sufficient quality for clarity?

Reviewer #1: The results are clearly and comprehensively presented, and they align well with the stated modelling framework and analysis plan described in the Methods section. The construction and interpretation of the ecological suitability index and the human spillover potential are logically sequenced and directly derived from the described input data, sub-models, and computational procedures. The results offer clear insight into regional variation in risk, supported by quantitative estimates and confidence intervals at national and sub-national levels. Figures and tables are of high quality, with clear legends and sufficient resolution to support interpretation. Spatial maps of host and vector distributions, ecological suitability, and spillover potential are particularly helpful and visually effective. The inclusion of validation results—comparing population-weighted suitability values between affected and unaffected Local Government Areas—adds credibility to the model's predictive utility. Overall, the results are appropriately detailed and well structured, with figures and supplementary materials enhancing the clarity and strength of the findings.

Reviewer #2: (No Response)

**Conclusions**

-Are the conclusions supported by the data presented?

-Are the limitations of analysis clearly described?

-Do the authors discuss how these data can be helpful to advance our understanding of the topic under study?

-Is public health relevance addressed?

Reviewer #1: The conclusions are well supported by the data presented and appropriately reflect the scope and findings of the analysis. The authors clearly describe the limitations of their modelling approach, including the static nature of the estimates, potential biases in ecological niche data, and uncertainties arising from limited empirical data on vector and host species distributions in Australia. These limitations are acknowledged transparently and constructively, with suggestions for future work to incorporate species-level models and temporal dynamics. The discussion effectively places the findings within the broader context of Japanese encephalitis virus emergence, offering clear insights into how this modelling framework advances understanding of ecological drivers and human spillover risk. Importantly, the public health relevance is strongly emphasized, particularly the value of identifying high-risk areas for targeted surveillance, vector control, and vaccination planning. The study provides a useful foundation for proactive interventions and risk mitigation in the face of ongoing environmental change.

Reviewer #2: (No Response)

**Editorial and Data Presentation Modifications?**

Reviewer #1: Overall, there are relatively minor revisions that, once addressed, will strengthen the manuscript's presentation and accessibility without requiring substantive changes to its analyses or conclusions.

Reviewer #2: (No Response)

**Summary and General Comments**

Reviewer #1: This manuscript presents a timely and well-executed modelling analysis of Japanese encephalitis virus (JEV) transmission risk and potential human spillover across Australia. By integrating host and vector ecology, vector competence, environmental suitability, and human population distribution into a high-resolution spatial framework, the authors provide a valuable and novel tool for informing public health interventions. The modelling approach is methodologically sound, appropriately validated, and well aligned with the study's stated objectives. The manuscript is clearly written and demonstrates a strong command of ecological modelling and public health relevance. The work is particularly important in the context of recent JEV outbreaks in non-endemic areas of Australia, and it contributes significantly to understanding the drivers of virus emergence and potential persistence.

Strengths of the study include its mechanistic framework, the thoughtful selection of species and ecological inputs, and the clear connection between ecological suitability and real-world public health applications. The limitations are acknowledged transparently and do not detract from the overall value of the findings. There are no concerns regarding research or publication ethics, and all data sources appear appropriately cited and openly accessible. Minor editorial and structural improvements are recommended to improve clarity, but no new experiments or analyses are required. With these revisions, the manuscript will be suitable for publication and will make a strong contribution to the literature on vector-borne disease ecology and preparedness.

Reviewer #2: The article makes a valuable contribution to the understanding of vector-host transmission dynamics and the spillover potential of Japanese encephalitis virus (JEV) in Australia, drawing on methodologies previously established in the literature. The approach adopted incorporates key ecological processes involved in transmission between vectors and hosts, as well as the density and distribution of the human population, to estimate exposure potential. The manuscript is well written, and the results are clearly presented. However, certain conceptual characteristics and properties of the constructed indices need to be more clearly defined and discussed.

PLOS authors have the option to publish the peer review history of their article (what does this mean? ). If published, this will include your full peer review and any attached files.

**Do you want your identity to be public for this peer review?** For information about this choice, including consent withdrawal, please see our Privacy Policy .

Reviewer #1: No

Reviewer #2: No

**Figure resubmission:**

**Reproducibility:**



---

## [Decision Letter · Decision Letter 1]

5 Nov 2025

Dear Dr. Sartorius,

We are pleased to inform you that your manuscript 'Ecological suitability of Japanese encephalitis virus in Australia: a modelling analysis of vector-host transmission dynamics to potential spillover in humans' has been provisionally accepted for publication in PLOS Neglected Tropical Diseases.

Best regards,

Maira Aguiar, PhD

Academic Editor

Qu Cheng

Section Editor

Shaden Kamhawi

co-Editor-in-Chief

Paul Brindley

co-Editor-in-Chief

Reviewer's Responses to Questions

**Key Review Criteria Required for Acceptance?**

**Methods**

-Are the objectives of the study clearly articulated with a clear testable hypothesis stated?

-Is the study design appropriate to address the stated objectives?

-Is the population clearly described and appropriate for the hypothesis being tested?

-Is the sample size sufficient to ensure adequate power to address the hypothesis being tested?

-Were correct statistical analysis used to support conclusions?

-Are there concerns about ethical or regulatory requirements being met?

Reviewer #1: (No Response)

Reviewer #2: (No Response)

**Results**

-Does the analysis presented match the analysis plan?

-Are the results clearly and completely presented?

-Are the figures (Tables, Images) of sufficient quality for clarity?

Reviewer #1: (No Response)

Reviewer #2: (No Response)

**Conclusions**

-Are the conclusions supported by the data presented?

-Are the limitations of analysis clearly described?

-Do the authors discuss how these data can be helpful to advance our understanding of the topic under study?

-Is public health relevance addressed?

Reviewer #1: (No Response)

Reviewer #2: (No Response)

**Editorial and Data Presentation Modifications?**

Reviewer #1: (No Response)

Reviewer #2: (No Response)

**Summary and General Comments**

Reviewer #1: (No Response)

Reviewer #2: The authors have adequately addressed the questions and incorporated the requested changes.

The manuscript is well written, and the results are presented clearly.

In this regard, I recommend the publication of the paper.

PLOS authors have the option to publish the peer review history of their article (what does this mean? ). If published, this will include your full peer review and any attached files.

**Do you want your identity to be public for this peer review?** For information about this choice, including consent withdrawal, please see our Privacy Policy .

Reviewer #1: No

Reviewer #2: No

---

## [Editor Report · Acceptance letter]

Dear Dr. Sartorius,

We are delighted to inform you that your manuscript, "Ecological suitability of Japanese encephalitis virus in Australia: a modelling analysis of vector-host transmission dynamics to potential spillover in humans," has been formally accepted for publication in PLOS Neglected Tropical Diseases.

Best regards,

Shaden Kamhawi

co-Editor-in-Chief

Paul Brindley

co-Editor-in-Chief
